# The Determinants of Sovereign Risk Premium in African Countries

**Jane Mpapalika [1,\*] and Christopher Malikane [2]**

[1]   Department of Knowledge Management and Innovation, Economic and Social Research Foundation (ESRF), Ursino Estates P.O. Box 31226, Tanzania

[2]   Department of Economics, School of Economics and Business Sciences, University of the Witwatersrand, Johannesburg 2000, South Africa; mlk83za@wits.ac.za

\*   Correspondence: jmpapalika@esrf.or.tz

**Abstract:** This paper investigates the determinants of the sovereign risk premium in African countries. We employ the dynamic fixed effects model to determine the key drivers of sovereign bond spreads. Country-specific effects are fixed and the inclusion of dummy variables using the Bai–Perron multiple structural break test is significant at a 5% level. For robustness, the time-series generalized method of moments (GMM) is used where the null hypothesis of the Sargan Test of over-identifying restrictions (OIR) and the Arellano–Bond Test of no autocorrelation are not rejected. This implies that the instruments used are valid and relevant. In addition, there is no autocorrelation in the error terms. Our results show that the exchange rate, Money supply/GDP (M2/GDP) ratio, and trade are insignificant. Furthermore, our findings indicate that public debt/GDP ratio, GDP growth, inflation rate, foreign exchange reserves, commodity price, and market sentiment are significant at a 5% and 10% level.

**Keywords:** sovereign risk/debt; risk premium; sovereign defaults; African countries

## 1. Introduction

This study investigates the determinants of sovereign risk premium in the selected African countries. As defined by Hilscher and Nosbusch (2010), sovereign risk premium is the difference between the interest rate on the sovereign bond and the interest rate on a risk-free U.S. Treasury bond of comparable maturity. According to Tkalec et al. (2014), the sovereign bond spread is the compensation to investors for default risk in case the realized welfare loss from default exceeds the expected welfare loss. Sovereign spreads are a proxy for the country risk premium which refers to the risk associated with the probability that a country will default on its debts. In general, Olabisi and Stein (2015), note that developing economies are subjected to persistent macroeconomic instabilities, and hence have a higher sovereign risk premium than advanced economies. From the sovereign risk management perspective, it is, therefore, important for policymakers to control the key drivers that influence a country's risk.

Understanding the sovereign risk premium determinants will assist policymakers in formulating sovereign risk management policies oriented towards minimizing sovereign default risks. For instance, Cassard and Folkerts-Landau (1997) argue that the objective of sovereign risk management is to ensure that the government's financing needs are met at the lowest possible cost of debt given a certain level of risk. Therefore, developing sound sovereign risk management policies to determine the key drivers of bond spread is essential to minimize risks and, in turn, achieve macroeconomic stability

of the wider economy. In addition, Korinek (2011) point out that the risk premium is an important factor in determining the choice between local currency debt and foreign currency debt, which is also crucial to policymakers. However, the main contribution of the study is to investigate the key drivers of sovereign risk premium in African countries. Our study builds on the works of Martinez et al. (2013), who employ a panel fixed effects model to examine the country risk premium determinants in emerging market economies. For robustness, we employ the Generalized Method of Moments with instrumental variables.

Despite the evidence of sovereign defaults in African economies, limited studies have been done on the determinants of sovereign risk premium in African countries. Studies such as Furceri and Zdzienicka (2012) have shown evidence of sovereign defaults in the selected African countries. However, Siklos (2011), points out that most of the studies focus on the determinants of bond spread in emerging market economies and developed economies. For instance, Kaminsky and Schmukler (1999), find that changes in the foreign interest rate increase the country spread and that the impact is high in countries with low credit ratings. Other studies, such as those by Mendonca and Nunes (2011), find that domestic variables are responsible for determining the risk premium. Hence, the gap that this study seeks to fill is that there are few studies focusing on the determinants of the sovereign risk premium in sub-Saharan Africa.

The rest of the paper is structured as follows: Section 2 reviews the literature; Section 3 presents the model; Section 4 provides the empirical methodology; Section 5 describes data; Section 5.1 summarizes stylized facts; Section 6 analyses empirical findings; and Section 7 concludes with some policy recommendations.

## 2. Related Literature

The literature on country risk premium determinants in sub-Saharan Africa is relatively scant, despite the empirical evidence of sovereign defaults. Most of the studies, such as that by Martinez et al. (2013), focus on the determinants of sovereign bond spreads in emerging market economies. This study uses quarterly data to estimate a panel fixed effects model on the seven Latin American countries, to investigate the determinants of bond spread. Their study finds inflation, terms of trade, the external debt, and international reserves to GDP ratio/GDP to be key drivers of sovereign bond spreads. Using panel data on emerging market economies, Baldacci and Kumar (2010) investigate the effect of fiscal deficits and public debt on long-term interest rates. They find that higher fiscal deficits lead to a significant increase in long-term interest rates. These self-fulfilling mechanisms suggest that higher fiscal deficits are likely to raise the sovereign risk premium, and hence, sovereign default risks.

According to the theory of fiscal insurance stipulated by Uribe (2006), under certain monetary and fiscal policy regimes, the sovereign risk default is inevitable. In fact, some of the monetary policy rules e.g., interest rate targeting and inflation targeting, are inconsistent with the central bank's objective of price stability. For instance, assume that a country is characterized by chronic fiscal deficits and the central bank's objective is price stability. Then, under this scenario, the probability of a country to default on its sovereign debt is high. More recently, Cuadra et al. (2010) realize that fiscal policy is still procyclical and that defaulting occurs more often in recessions. If the government ran a deficit budget, it has the option of financing this deficit by borrowing from abroad, though at a high interest rate. As observed by Bi (2012), if taxes are more distortionary, it will be more likely that the government will raise its foreign borrowing, and hence, a high likelihood to sovereign default risk.

There has been inconclusive evidence regarding the determinants of sovereign risk premium in developing countries. While some studies, such as those by Baldacci et al. (2011), Iara and Wolff (2014), indicate that domestic fundamentals are responsible for changes in the sovereign risk premium. For instance, Tkalec et al. (2014) demonstrate that an increase in current account deficit raises the level of sovereign risk premium. In addition, De Grauwe and Ji (2012) point out that a high ratio of public debt/GDP increases the debt-servicing burden, which in turn raises debt-servicing costs, along with

the likelihood of a sovereign default. Other studies find the market sentiment to be the key driver of sovereign bond spread, in particular, global risk aversion. As noted by Audzeyeva and Schenk-Hoppé (2010), the investor's sentiments significantly affect the bond spread. Similarly, Siklos (2011), highlight that the degree of investor's risk aversion is an important factor in explaining bond spread variations.

The economic literature has identified that most of the variations in the sovereign bond spread are explained by GDP growth. Findings by Sturzenegger (2004), prove that an increase in debt burden is associated with low GDP growth of about 0.6–2.2%. This implies that an improvement in the economic performance of a country contributes to macroeconomic stability. As a result, the probability of sovereign default occurring is minimal. A high level of GDP growth raises the country's ability to service its debt burden, which leads to a reduction in the level of risk premium (Borensztein and Panizza 2009; Maltritz and Molchanov 2013). Other studies, e.g., Galindo et al. (2003) and Gumus (2011), have argued that the contractionary effects of the exchange rate depreciation are associated with currency mismatches and sudden stops, which may lead to financial instability. According to Calvo and Reinhart (2002), this is the main reason many developing countries exhibit the "fear of floating", and therefore, adopt managed floats or soft pegs.

Other studies such as that of Baldacci et al. (2011) find that inflation is significant in influencing the sovereign risks. As described by Martinez et al. (2013), a high inflation rate may be attributed to the monetization of the fiscal deficit and the need for higher interest rates. Hence, Min et al. (2003), illustrate that an increase in inflation rate signaling macroeconomic instability will consequently raise the sovereign risk premium. However, the default risk will be high in an inflation-targeting economy if the government will not raise the short-term interest rate through the central bank to curb inflationary tendencies. The expected sign of inflation on the risk premium is positive.

Foreign exchange reserve/GDP is also an important factor in determining the level of risk premium. The accumulation of foreign exchange reserves should reduce the country's risk. Hence, the hypothesized sign is negative, as shown by (Tkalec et al. 2014). As noted by Bellas et al. (2010) and Martinez et al. (2013), a low ratio of foreign reserves/GDP translates into a high likelihood of sovereign default and liquidity risks. In addition, Ouyang and Rajan (2014), stress that the accumulation of foreign exchange reserves acts as an insurance against sovereign defaults.

Global factors such as global liquidity, risk appetite, and contagion effects are important in determining the country risk. This is in line with findings by Gonzalez-Rozada and Levy-Yeyati (2008) that the level of risk reflects investors' risk aversion. Moreover, Kamin and Kleist (1999), find that bond spreads are mainly explained by the shifts in market sentiments, rather than by variations in fundamentals. Similarly, Maltritz and Molchanov (2013), conclude that market sentiments play an important role in determining the sovereign bond spread over time. However, it is generally argued that developing countries are riskier than advanced economies. The expected sign of market sentiment on the country risk is positive.

## 3. The Model

Following Ferrucci (2003) and Comelli (2012), a conventional approach to sovereign bond yield is the assumption that the yield spread is a function of the likelihood that a default will occur and the welfare loss following a default. This likelihood of a default occurring depends on a set of macroeconomic fundamentals, e.g., the public debt/GDP ratio, inflation, real exchange rate, and fiscal deficits. Other studies, such as those by Gonzalez-Rozada and Levy-Yeyati (2008), Kamin and Kleist (1999), Maltritz and Molchanov (2013) conclude that market sentiments play an important role in determining the sovereign bond spread over time. In addition, Uribe (2006), makes an important observation regarding the probability of default risks in that, under certain monetary and fiscal policy regimes, the sovereign risk default is inevitable. This is attributed to the inconsistent and conflicting objectives of fiscal and monetary policy regimes.

Therefore, Jankovic and Zivkovic (2014), declare that the bond yield spread is negatively correlated with foreign debt sustainability. As developed by Edwards (1984), the sufficient condition for the optimal share of the investor's funds is given by:

$$\left(1 + r_f\right) = p_w + (1 - p)\left(1 + r^L\right) \tag{1}$$

where $r_f$ is the risk-free interest rate, $p$ is the default probability of the debtor country, $p_w$ is the debtor's payment to the investor in the case a default occurs, and $r^L$ is the rate of return on investment. Then, Bellas et al. (2010), suggest that the probability of default on emerging market sovereign bonds and the risk-free interest rate have the same maturity. In the long-run, however, the probability of sovereign default is constant. Hence, the bond spread over the U.S. Treasury bond, $s$, is given by:

$$s = r^L - r_f = \frac{p}{1 - p}\left(1 + r_f\right) \tag{2}$$

With respect to Edwards (1984), the probability of default in a logistic form is specified as:

$$p = \frac{\exp\left(\sum\limits_{j=1}^{J} \beta_j x_j\right)}{1 + \exp\left(\sum\limits_{j=1}^{J} \beta_j x_j\right)} \tag{3}$$

By combining Equations (2) and (3), and after taking the logarithm, the sovereign risk premium is provided by:

$$\log s = \log\left(1 + r_f\right) + \sum\limits_{j=1}^{J} \beta_j x_j + \varepsilon_{it} \tag{4}$$

Equation (4) shows that the sovereign bond spread at time $t$ is affected by the country's fundamentals, $x_j$, and the risk-free interest rate, $r_f$. Expressing Equation (4) in logs will generate the following log-linear specification with fixed individual effects:

$$\log(embig_{it}) = \alpha_0 + \sum\limits_{j=1}^{n} \alpha_j x_{ijt} + \gamma_j + \lambda_t + \varepsilon_{it} \tag{5}$$

where $embig_{it}$ is the sovereign bond yield spread for country $i$ in time $t$, $x_{ijt}$ is a vector of independent variables that includes country-specific and global factors, $\mu_i$ is the country-specific fixed effects, and $\varepsilon_{it}$ are the error terms, which are independently and identically distributed.

## 4. Empirical Methodology

### 4.1. Panel Unit Root Tests: LLC and IPS

It is important to carry out the unit root tests in order to determine the order of integration. Moreover, the regression of non-stationary time series data is likely to give spurious results. As noted by Strauss and Yigit (2003), panel unit root tests, address the low power problem in an individual-equation Augmented Dickey–Fuller Test by averaging the t-statistics across the panel and assumes that the disturbance terms are normally distributed. This study will use panel unit root tests developed by Levin et al. (2002) and Im et al. (2003) to determine the order of integration of the underlying variables of interest. The null hypothesis for these unit root tests is that all panels contain a unit root, I(1), except the null hypothesis for the Hadri Unit Root Test that assumes all panels are stationary, I(0).

The Im et al. (2003), based on the Dickey–Fuller procedure specifies each cross-section with individual effects and no time trend as follows:

$$\Delta y_{it} = \alpha_i + \gamma_i y_{i,t-1} + \sum_{j=1}^{p_i} \chi_{ij} \Delta y_{i,t-j} + \varepsilon_{it} \tag{6}$$

where $i = 1, \dots, N$ and $t = 1, \dots, T$.

The Im, Pesaran and Shin uses separate unit root tests for the $N$ cross-section units. After estimating the separate Augmented Dickey Fuller regressions, the average of t-statistics for $p_1$ from the individual ADF regressions, that is:

$$\bar{t}_{NT} = \frac{1}{N} \sum_{i=1}^{N} t_{iT}(p_i \beta i)$$

The *t*-bar statistic is then standardized and converge to the standard normal distribution as $N$ and $T$ tends to infinity. As shown by Im et al. (2003), the *t*-bar statistics is robust when $N$ and $T$ are small.

*4.2. Residual-Based Panel Cointegration Test*

If the null hypothesis of a unit root test is not rejected, then the cointegration test can be performed to determine the long-run relationship in the underlying variables. In a conventional way, cointegration refers to the set of variables that are individually integrated of order one, some linear combination of these variables can be described as stationary, I(0). Earlier works of Pedroni (2004), investigate the properties of residual-based tests for the null of no cointegration for both homogeneous and heterogeneous panels. An interesting observation by McCoskey and Kao (1998) suggests that testing the null hypothesis of cointegration rather than the null hypothesis of no cointegration is interesting, especially in applications, where cointegration is forecasted a priori by economic theory.

However, Pedroni (1999), note that residual-based tests are more robust even in small samples although they cannot determine the number of co-integrated relationships. Our study uses the Pedroni cointegration test suggested by Pedroni (1999). The procedures proposed by Pedroni make use of the estimated residual from the hypothesized long-run regression in its general form, which is given by:

$$y_{it} = \alpha_i + \delta_i t + \beta_i x'_{it} + e_{it} \tag{7}$$

where $i = 1, \dots, N$ over time periods $t = 1, \dots, T$. The variables $y_{it}$ and $X_{it}$ are assumed to be integrated of order one, denoted by I(1), and under the null of no cointegration, the residual will also be I(1). The parameters $\alpha_i$ and $\delta_i$ allow for the possibility of specific fixed effects and deterministic trends. The slope coefficient $\beta_i$ is also allowed to vary so that the cointegrating vectors may be heterogeneous across the panel.

Pedroni (1999) suggests the heterogeneous group mean panel test statistics for panel cointegration asymptotic distribution of individual t-statistics is given by:

$$\frac{X_{N,T} \mu \sqrt{N}}{\sqrt{v}} \Rightarrow N(0,1)$$

where $X_{N,T}$ correspond to test statistics, while $\mu$ and $v$ are the mean and variance of each test, respectively.

If the variables are cointegrated, then, we proceed to estimate Equation (8) using the panel dynamic fixed effects method. We build our model on the works of Martinez et al. (2013), who investigate the determinants of sovereign risk premium in emerging market economies using panel fixed effects. Panel fixed effect models control for unobservable individual and time-specific effects. According to Csonto and Ivaschenko (2013), a panel model takes into account the individual dimension and the time dimension. As noted by Judson and Owen (1999), the fixed effects model is

more robust than the random effects model, because it also controls endogeneity problems. However, the general model of bond spread determinants for country $i$ at time $t$ is given by the following dynamic fixed effects model:

$$y_{it} = \gamma y_{i,t-1} + \sum_{j=1}^{n} \alpha_j x_{ijt} + \gamma_j + \lambda_t + \varepsilon_{it}, \; For \; i = 1 \ldots N \text{ and } t = 1 \ldots T \tag{8}$$

where $y_{it}$ is the sovereign risk premium measured by the interest rate on the sovereign bonds and the 10-year US Treasury bond, which is the benchmark and default-free. Most of the studies use the JP Morgan Emerging Markets Global Bond Index, which is the most comprehensive emerging markets benchmark. Despite the fact that this index incorporates bonds issued in emerging market economies, Asia, Africa, and Europe, it seldom includes all the low-income countries. According to Ebner (2009), some European countries are not included in the JPMBIG index. On the other hand, this benchmark index includes African countries, such as Egypt, South Africa, Nigeria, and Morocco.

However, Ferrucci (2003) and Ebner (2009), use the emerging market benchmark. The vector of independent variables that includes country-specific and global factors is represented by $\chi_{it}$. As shown by Comelli (2012) and Martinez et al. (2013), sound country-specific factors are associated with lower bond spreads. An appreciation of the real exchange rate decreases the country risk premium. Therefore, the expected sign for the exchange rate is negative. In addition, some studies, such as those by Baek et al. (2005); Bellas et al. (2010), and Siklos (2011), find that high GDP growth, greater amounts of reserves and an increase in money supply lower the sovereign bond spread. Hence, this study expects a negative impact on sovereign bond spreads. Other studies, such as Gylfason (1991), point out that higher inflation rates contribute to higher sovereign default risks.

The risk premium is high in heavily indebted countries because of the worsening of fiscal deficit and an increase in investor's sentiments. The hypothesized sign is positive. In addition, the increase in the investors' risk aversion raises the risk premium. In this sense, Bernoth et al. (2012) are of the opinion that in times of uncertain economic conditions, rational investors move to less risky assets, which, in turn, raises the bond spread. In order to control for global factors, the market sentiment is proxied by the consumer's sentiment. Some studies, such as those by Aristei and Martelli (2014), have used direct proxies for market sentiment based on consumer confidence. $\zeta_i$ represents the individual country-specific effects, $\gamma_t$ the time effect, and $\varepsilon_{it}$ the error term. This study has also considered two dummy variables in order to capture the effects of the debt crisis. The individual country-specific effect is denoted by $\gamma_i$ and the time-varying unobservable time effect is denoted by $\lambda_t$. The error term is represented by $\varepsilon_{it}$. In addition, we investigate the structural breaks by adding dummy variables.

As a means of robustness, this study uses the system generalized method of moments (GMM) established by Arellano and Bover (1995) and Blundell and Bond (1998). It enhances the difference GMM estimator by creating an additional assumption that first differences of instrument variables are uncorrelated with the fixed effects. The two-step estimator is more efficient than the one-step system GMM and performs better than the differenced GMM and within-group estimators. To address endogeneity and simultaneity bias issues, the explanatory variables are lagged to more than two periods which are used as instruments. If the moment conditions are valid, Blundell and Bond (1998) show that, in Monte Carlo simulations, the GMM-system estimators perform better than the GMM-difference estimator.

We test the validity of the moment conditions by using the Sargan test of over-identifying restrictions. Further tests include an Autoregressive (AR) test to check if the differenced error terms are first and second order serially correlated. The principle of this method is to choose instruments, which satisfy a set of orthogonality conditions, whereby the number of instruments ought not to exceed the number of panels. The order condition requires that the number of instruments must be at least equal to the number of endogenous components, such that $r < K$. Roodman (2009), note that using many moment conditions can improve efficiency but makes the GMM inferences inaccurate.

Bound et al. (1995) argue that the use of an instrumental variable in the case of a weak relationship between instruments and the endogenous explanatory variable causes large inconsistencies in the estimated instrumental variables.

## 5. Data Description

This study employs annual data covering the period 1971 to 2011 for 10 African countries. The countries selected include Burundi, Egypt, The Gambia, Kenya, Mauritania, Nigeria, Sierra Leone, Swaziland, South Africa, and Zambia. Data is sourced from the International Monetary Fund (IMF)'s International Financial Statistics, the World Bank development indicators; the UNCTAD statistics, the Federal Reserve Bank of St. Louis, and each country's central bank. The country risk premium is measured by the difference between the interest rate on the sovereign bond denominated in US Dollars and the interest rate on the 10-year US Treasury bond, which is the benchmark and default-free. Data on the interest rate of sovereign bond in African countries are obtained from the country's central bank and the Federal Reserve Bank of St. Louis. The 10-year US T-bond is sourced from the Federal Reserve Bank of St. Louis. In addition to the JPEMBIG index, Comelli (2012), point out that this index includes Egypt and South Africa whereas, Bunda et al. (2009), show that the index includes Morocco as well.

The selection of the countries is based on the fact that these countries have a domestic debt market where information on sovereign bonds is available. As noted by Christensen (2005), who investigated the domestic debt markets in sub-Saharan Africa, their choice of countries was limited to non-CFA Franc countries, since CFA Franc countries, until very recently, did not have any domestic debt markets. Among the non-CFA countries, Angola, Botswana, the Democratic Republic of the Congo, Mozambique, and São Tomé and Príncipe did not have domestic government debt markets at the time of collection, that is, in 2005. In cases where these were insufficient, central bank reports or IMF country desk economists helped to fill their gaps.

Inflation is measured by the consumer price index. The exchange rate is defined as the value of domestic currency per unit of foreign currency. GDP growth rate is the real GDP at 2005 prices. Total reserves/GDP is the ratio of reserves to GDP. M2/GDP is the ratio of money supply to GDP where M2 includes M1 with all savings and time-related deposits. The fiscal deficit is measured by the public debt/GDP ratio. Current account/GDP is the sum of the balance of trade, net factor income from abroad, and net cash transfers to GDP growth rate. In order to control for global factors, the proxy for market sentiment is given by the consumer's sentiment. Studies such as Aristei and Martelli (2014) have used direct proxies for market sentiment based on consumer confidence.

### 5.1. Stylized Facts

Table 1 reports the correlations between the sovereign risk premium and its determinants in African countries. The correlation results show that most of the series are highly correlated with each other. For instance, the correlation between M2/GDP and exchange rate is 0.89; M2/GDP and current account/GDP is 0.85; real GDP and public debt/GDP is 0.70; and the exchange rate and current account/GDP is 0.84. In this case, some insignificant variables will be dropped to avoid the problem of multicollinearity. Most of the studies, e.g., those by Comelli (2012), find a negative correlation between the bond spread and its determinants. This implies that improvement in country-specific fundamentals, such as GDP growth, is associated with low sovereign risk premium. Our results show that there is a negative correlation between the GDP growth rate and the bond spread. The increase in economic growth raises the tax revenue used for servicing debt and, therefore, there is a decline in default risks. This is in line with studies by Siklos (2011) and Maltritz and Molchanov (2013), who show that the bond spread reduces in response to an improvement in the GDP growth rate.

**Table 1.** Correlation analysis.

| Variables | Risk Premium | Reserves/GDP | M2/GDP | Inflation Rate | Real GDP | Exchange Rate | Public Debt/GDP | Consumer Sentiment | Current Account/GDP Ratio |
|---|---|---|---|---|---|---|---|---|---|
| Risk Premium | 1 | | | | | | | | |
| Reserves/GDP | −0.176 ** | 1 | | | | | | | |
| M2/GDP | 0.648 ** | −0.416 ** | 1 | | | | | | |
| Inflation rate | 0.165 ** | −0.032 *** | 0.186 ** | 1 | | | | | |
| Real GDP | −0.216 ** | 0.149 ** | 0.163 ** | −0.152 ** | 1 | | | | |
| Real exchange rate | −0.723 ** | −0.518 ** | 0.893 ** | 0.146 ** | 0.02 *** | 1 | | | |
| Public debt/GDP | 0.006 *** | −0.637 ** | 0.088 ** | 0.09 ** | 0.702 ** | 0.18 ** | 1 | | |
| Consumer sentiment | 0.203 ** | −0.229 ** | −0.355 ** | −0.267 ** | −0.17 ** | −0.30 ** | 0.400 ** | 1 | |
| Current account/GDP | −0.431 ** | −0.44 ** | 0.85 ** | 0.009 *** | 0.057 *** | 0.842 ** | 0.134 ** | −0.223 ** | 1 |

** and *** denotes statistical significance at 5% and 10% level.

There is a negative correlation between foreign exchange reserves/GDP and bond spread. If the level of reserve accumulation declines, then the likelihood of a sovereign default is high. These results are consistent with Tkalec et al. (2014), who show that the accumulation of foreign exchange reserves raises a country's liquidity level, thereby lowering the sovereign bond spread. In practice, foreign exchange reserves are accumulated to raise a country's ability to meet its external debt obligations. More recently, Maltritz and Molchanov (2013), find that foreign reserves/GDP growth is negatively correlated with the sovereign risk premium. In addition, Bellas et al. (2010) and Martinez et al. (2013) argue that a country's foreign exchange reserves can be used to service foreign debt and fix the exchange rate. Hence, the hypothesized sign is negative.

There is a positive correlation between the inflation rate and the sovereign bond spread. A high inflation rate reflects macroeconomic instability and consequently, a rise in the country risk premium. As noted by Calvo et al. (1996), an effective inflation-targeting policy will reduce macroeconomic instabilities by managing inflationary tendencies, which may lead to capital outflows. On the other hand, Baldacci et al. (2011), observe that a rise in the inflation rate, which is attributed to the adoption of an expansionary monetary policy, contributes to a high likelihood of sovereign risks. Other studies, such as those by Martinez et al. (2013), find that the inflation rate is significant at 1% and exhibits the expected positive sign. This means that the bond spread will increase in response to inflationary pressure. More recently, Tkalec et al. (2014) show that sovereign risk premium increases in response to higher inflation in the long run.

The public debt/GDP ratio is positively correlated to the bond spread. It clearly suggests that an increase in public debt/GDP ratio contributes to a rise in the level of the sovereign risk premium. This is consistent with Bernoth et al. (2012), who observe a positive relationship between bond yield spreads and fiscal variables. Similarly, Tkalec et al. (2014), find that the country risk premium increase in response to a higher total debt to GDP ratio, which is attributed to the high cost of debt servicing. In addition, De Grauwe and Ji (2012), affirm that the debt to GDP ratio has a priori expected positive sign towards the bond spread. As noted further by Bi (2012), when the level of public debt is beyond the fiscal limit, sovereign default is likely to occur. On the other hand, the current account balance/GDP ratio is negatively correlated to the sovereign risk premium. According to Özatay et al. (2009) and Martinez et al. (2013), this means that the increase in current account balance/GDP that is current account surplus reduces the probability of sovereign defaults.

Furthermore, market sentiments and bond spreads are positively correlated, which implies that the increase in the investor's risk aversion raises the sovereign risk premium. This is in contrast with Gonzalez-Rozada and Levy-Yeyati (2008), who observe that the emerging market bond spread is negatively correlated with the international risk appetite. Other studies, e.g., Dailami et al. (2008), reveal that an increase in global risk aversion increases the sovereign bond spreads through the contagion effect. An increase in market sentiments is reflected by the improvement in the sovereign credit rating. As shown by Ebner (2009) and Zinna (2013), emerging market sovereign risk is driven by the degree of investors' risk aversion.

## 6. Empirical Results

### 6.1. IPS and LLC Unit Root Tests

Table 2 presents the panel unit root tests using studies by Levin et al. (2002) and Im et al. (2003). The optimal lag length selection is chosen based on the Schwarz Information Criterion (SIC) and the optimal bandwidths of LLC; IPS unit root tests are determined based on the Newey–West criterion. Unit root test procedures are designed to investigate the stationarity of each individual series in the panel. Most of the variables, e.g., the exchange rate, M2/GDP ratio, public debt, foreign exchange reserve, and trade accept the null hypothesis of a unit root. However, after first differencing these series, they become stationary at the 5% significance level, with individual linear trends. This is consistent with the findings of Martinez et al. (2013). Other series such as the GDP growth rates, inflation rates,

market sentiment, sovereign risk premium, and commodity prices are stationary. We then carry out the Pedroni residual-based cointegration after stationarity of the series.

**Table 2.** Panel unit root test.

| Variables | Level (Without Trend) | | Difference (With Trend) | |
| :---: | :---: | :---: | :---: | :---: |
| | Levin, Lin, and Chu | Im, Pesaran, and Shin | Levin, Lin, and Chu | Im, Pesaran, and Shin |
| GDP | −22.38 ** | −21.06 ** | −19.66 ** | −19.63 ** |
| Exchange rate | 1.86 *** | 5.51 *** | −9.03 ** | −5.03 ** |
| Inflation rate | −17.57 ** | −14.24 ** | −16.85 ** | −12.71 ** |
| M2/GDP | 0.01 *** | 2.05 *** | −25.03 ** | −29.26 ** |
| Public debt/GDP | | | −23.00 ** | −24.34 ** |
| Foreign exchange reserves | | | −10.27 ** | −13.41 ** |
| Risk premium | −2.29 ** | −2.79 ** | | |
| Market sentiment | −6 ** | −7.8 ** | | |
| Trade/GDP | | | −11.03 ** | −18.77 ** |
| Commodity price | −3.09 ** | −7.29 ** | | |

(i) The null hypothesis of no unit root is rejected if *p*-value < 0.05. (ii) ** and *** denotes 5% and 10% *** significance level.

## 6.2. Panel Residual-Based Cointegration Test

Table 3 reports the residual-based panel co-integration results. According to Pedroni (1999) and Kao (1999), the advantage of a residual-based test over the maximum-likelihood based co-integration test is that they are robust, even in small samples. However, Banerjee (1999), points out that they are unable to determine the number of co-integrated relationships. Our test shows that the null hypothesis of no co-integration is strongly rejected in the series under investigation. Since there are co-integrating relationships between the bond spread and its determinants, we proceed to estimate the key drivers of sovereign bond spread in African countries using the Dynamic Fixed Effects model and the Generalized Method of Moments for robustness.

**Table 3.** Pedroni residual-based cointegration test.

| Variable | Without Deterministic Trend | With Deterministic Trend |
| :---: | :---: | :---: |
| Sovereign risk premium | −2.66 ** | −2.38 ** |

(i) ** and *** denotes 5% and 10% significant levels. (ii) Null hypothesis is rejected if *p*-value < 0.05.

## 6.3. Dynamic Fixed Effects Estimation

Table 4 reports the estimation results for dynamic fixed effects on the sovereign risk premium determinants. The specification allows country-fixed effects to account for cross-country heterogeneity that affects all countries. As pointed out by Bordo et al. (2010), a dynamic fixed effects estimator is robust to heteroscedasticity and serial correlation in the individual dimension. In Table 4, column (1) is the baseline specification, which includes all the variables of interest, where the exchange rate and M2/GDP are insignificant. Column (2) shows the insignificant variables. Column (3) incorporates the dummy variables. It turns out that the inclusion of time dummies affects the significance of trade/GDP ratio, where it becomes insignificant. Our results are consistent with Martinez et al. (2013), who confirm that the exchange rate is insignificant in emerging market economies. In contrast, Ebner (2009), find that the exchange rate is highly significant in African countries.

**Table 4.** Dynamic Fixed Effects estimation results.

| Variables | (1) | (2) | (3) |
|---|---|---|---|
| $Risk_{t-1}$ | 0.65(0.04) ** | 0.72(0.03) ** | 0.63(0.04) ** |
| GDP | 0.07(0.01) ** | 0.04(0.01) ** | 0.07(0.01) ** |
| Exchange rate | −0.06(0.09) | | |
| Inflation rate | 0.04(0.01) ** | 0.03(0.01) ** | 0.04(0.01) ** |
| M2/GDP | 0.06(0.06) | | |
| Public debt/GDP | 0.03(0.007) ** | 0.01(0.007) ** | 0.03(0.007) ** |
| Foreign exchange reserves | −0.06(0.01) ** | −0.06(0.01) ** | −0.06(0.017) ** |
| Market sentiment | 0.19(0.09) ** | 0.21(0.10) ** | 0.17(0.09) *** |
| Trade/GDP | −0.07(0.05) ** | −0.11(0.05) ** | |
| Commodity price | −0.11(0.03) ** | −0.15(0.03) ** | −0.12(0.03) ** |
| D1979 | | | 0.16(0.04) ** |
| D1980 | | | 0.16(0.04) ** |
| D2001 | | | 0.17(0.04) ** |
| D2007 | | | 0.20(0.04) ** |
| D2008 | | | 0.13(0.04) ** |
| $R^2$ | 0.64 | 0.61 | 0.67 |
| Durbin–Watson statistics | 1.79 | 1.65 | 1.83 |
| Observations | 344 | 344 | 344 |
| Prob(F-statistics) | 0.0000 | 0.0000 | 0.0000 |

Notes: (i) ** and *** indicates statistical significant at 5% and 10%, respectively. (ii) Standard errors in parentheses.

The results of this study show that a 1% increase in GDP growth, contributes to a decline in the sovereign risk premium by 0.65%. The coefficient on the GDP growth is statistically significant at 5% and displays the expected negative sign. In contrast, Martinez et al. (2013) find that the GDP growth and the public debt/GDP ratio are statistically insignificant, although they present the correct sign. However, our findings are in line with those of Baek et al. (2005), Siklos (2011), Maltritz and Molchanov (2013), who note that high growth rates reduce the sovereign default risk. This means that the increase in the general economic performance of a country raises the macroeconomic stability in the fiscal variables hence, lowering the probability of a sovereign default. For instance, Guillaumont et al. (1999), indicate that macroeconomic fluctuations lower economic growth in African countries. Intuitively, the coefficient on real GDP growth shows that low-growth countries are penalized when issuing bonds as they tend to do so at higher spreads.

The coefficient on the inflation rate is significant and has a positive sign. This implies that a 1% increase in the inflation rate leads to an increase in the sovereign risk premium by 4%. This is in contrast with the findings of Mendonca and Nunes (2011) and Maltritz (2012), where inflation is statistically insignificant in influencing the level of risk premium. However, our findings are consistent with Baldacci et al. (2011), who find that a rise in sovereign bond spread leads to a high inflation rate. In practice, there is a strong link between inflationary tendencies and sovereign defaults. For instance, Reinhart and Rogoff (2011) singled out Zimbabwe, which has weak fiscal and monetary institutions and is significantly exposed to high inflation and the risk of a sovereign default. Using an inflation threshold of 20%, their study finds that Zimbabwe's annual inflation rate exceeds 500%. On the other hand, Eichengreen et al. (2002), cites that inflation-forecast targeting countries like South Africa and Chile, which have strong monetary, fiscal, and financial institutions, normally experience a lower sovereign risk premium.

However, further analysis indicates that Egypt, Ghana, and South Africa are effectively implementing inflation targeting as a monetary policy rule, so as to keep inflation in check. Some studies, such as those by Aron and Aron and Muellbauer (2005), find that since the Reserve Bank of South Africa implemented inflation targeting to manage its inflation in the 2000s, monetary policy credibility increased, and the GDP growth improved, as a result, it lowers the sovereign risks. Similarly, Kyereboah-Coleman (2012), confirms that inflation targeting is successful in managing inflation, lowering the exchange rate pass-through, and improving the credibility of the monetary policy in Ghana. On the contrary, Malikane and Mokoka (2012) find that since the implementation

of inflation targeting, monetary policy is not credible when managing inflation. Countries with high inflation rates do not need to issue large amounts of debt, as the inflation tax is a major source of government revenue. For instance, Claessens et al. (2007), show that high inflation is associated with macroeconomic instability, and occasionally, with general government defaults.

A 1% increase in the public debt/GDP ratio raises the sovereign risk premium by 0.03% and the coefficient on the public debt/GDP is highly significant. This suggests that as the public debt/GDP ratio increases, the debt-servicing burden also rises. In turn, indebtedness affects the sovereign's creditworthiness, which leads to a high level of country risks. This observation is in contrast with the findings by Bernoth et al. (2012) and Bernoth and Erdogan (2012), where the coefficients on the debt variable are insignificant. Importantly, it is advisable that a low public debt/GDP ratio is desirable so as to maintain macroeconomic stability in the fiscal variables. The study findings further confirm the existence of the "debt-overhang hypothesis" in Sub-Saharan Africa. The fact that most African economies with the exception of South Africa have high shares of foreign currency liabilities reflects the presence of poor sovereign debt risk management policies and weak institutions.

A 1% reduction in foreign exchange reserves leads to an increase in the sovereign risk premium by 0.06% and is significant at 5% level. The accumulation of foreign exchange reserves is associated with the increase in liquidity, which, in turn, reduces the bond spread. The results of this study contrast with those of Heinemann et al. (2014), who find that the foreign exchange reserve is insignificant in all specifications. However, our findings are consistent with those of Tkalec et al. (2014) and Maltritz and Molchanov (2013) who demonstrated the decline in the sovereign bond spread following an increase in reserve accumulation. However, Gelos et al. (2011) and Olabisi and Stein (2015) reserve accumulation as a monetary policy instrument for providing liquidity acts as an insurance against sovereign defaults. Monetary authorities usually use foreign exchange reserves when there is high exchange rate volatility, in order to fix the exchange rate. Another motive identified by Jeanne and Ranciere (2011) is that countries hold reserves covering three months' worth of imports.

The coefficient on the market sentiment has the expected positive sign and is statistically significant at 5%. This indicates that the bond spread increases by 0.19% (19 basis points) in response to a 1% reduction in the investor's risk aversion. These results are in line with Bernoth et al. (2012) who note that the increase in the bond spread is caused by the rise in the investors' risk aversion. More recently, Tkalec et al. (2014) confirm that the sovereign risk premium rises in response to high market sentiments. In addition, Cuadra et al. (2010) observe that when investors perceive the likelihood of sovereign defaults, country risks will start rising. In this case, a rise in investor's sentiments could be interpreted as a lack of sound risk management strategies to maintain macroeconomic stability. However, our findings also show that African countries are significantly exposed to commodity price shocks. As investigated by Claessens and Qian (1992), African economies depend heavily on the undiversified primary export commodities, which are highly volatile.

Diagnostic checks, such as the Durbin–Watson Test statistics, show no autocorrelation, and the R-squared shows that the model fits our data well. The R-squared increases due to additional explanatory variables. Similarly, Hilscher and Nosbusch (2010) find that adding country-specific variables leads to a substantial increase in the adjusted R-squared. The inclusion of dummy variables is significant in all the specifications.

### 6.4. Robustness Check: GMM

Table 5 reports the time-series analysis based on the GMM model to determine the key drivers of sovereign risk premium in sub-Saharan Africa. Our results show that the exchange rate, M2/GDP, and trade are insignificant in influencing the sovereign risk premium. This is consistent with the findings of Martinez et al. (2013). We perform the Hansen J-Test of over-identifying restrictions for OIR proposed by Sargan (1958), which tests for the overall validity and relevance of the instruments used. In addition, we carry out the Arellano–Bond Test of no autocorrelation in the error terms. The null hypothesis for the Hansen J-Test is that the instruments are valid and relevant. While, the null hypothesis of the

Arellano and Bond autocorrelation tests of first and second order AR (1) and AR (2), is that there is no autocorrelation. Our results do not reject the null hypothesis of these tests. To address endogeneity issues, Blundell and Bond (1998) suggest the use of instrumental variables by lagging the explanatory variables to two or more lags.

**Table 5.** Generalized Method of Moment estimation results.

| Variables | Inflation Rate | GDP | Public Debt/GDP | Reserves | Sentiment | Trade/GDP | Commodity Price |
|---|---|---|---|---|---|---|---|
| Burundi | 0.14(0.04) ** | | 0.02(0.01) *** | | | | −0.5(0.18) ** |
| Egypt | 0.12(0.07) ** | | | | 3.44(0.57) ** | | |
| Gambia | −0.31(0.13) ** | | | | | | |
| Kenya | | 0.97(0.5) *** | | | 4.18(2.24) *** | | |
| Mauritania | | | | −0.31(0.15) ** | 0.99(0.34) ** | | |
| Nigeria | | −0.26(0.09) ** | | 0.86(0.21) ** | | | |
| Sierra Leone | 0.47(0.10) ** | | | 0.97(0.18) ** | 1.64(0.87) *** | | |
| Swaziland | | | | −0.19(0.05) ** | | | −0.32(0.09) ** |
| South Africa | | | | | | | |
| Zambia | 0.73(0.11) ** | 0.29(0.10) ** | | | | | |

(i) ** and *** indicates statistical significant at 5% and 10%, respectively. (ii) Standard errors in parentheses. (iii) Instrumental variables include lagged risk premium, lagged exchange rate, lagged M2/GDP, lagged GDP growth, lagged trade/GDP, lagged inflation rate, lagged public debt/GDP, lagged market sentiment, and lagged commodity price (instrumented to one lag).

An important rule of thumb when choosing the number of instruments is that the number of instruments should be at least or equal to the number of regressors, in order not to violate the order condition and limit the number of lags of endogenous variables to $t − 1$ and $t − 2$. However, for instruments to be valid, they should be strictly exogenous. According to Stock et al. (2002), weak instruments arise when the instruments are weakly correlated with the endogenous variables. The model uses lagged values of the endogenous variable and differences of independent variables as valid instruments. In addition, the exchange rate and M2/GDP ratio are used as instruments. The instrumental variables used are correlated with the endogenous variable such that there is reverse causality. For instance, Devereux and Lane (2003), demonstrate that there is potential endogeneity between risk premium and the exchange rate, as well as the M2/GDP ratio. However, the seven fundamental variables and five time dummies are taken as strictly exogenous. According to Kennedy (1998), the instrumental variable technique provides consistent estimates.

The coefficient estimate on the inflation rate is significant at 5% level in Burundi, Egypt, Gambia, Sierra Leone, and Zambia. This is in line with the findings by Ferrucci (2003), who respectively suggest that macroeconomic variables such as domestic inflation rates are significant determinants of sovereign risk spreads. However, several studies, e.g., Fischer (1993) and Barro (1999), show a negative correlation between inflation and growth, because countries that experience poor growth performance normally have high inflation rates. Again, they respond to growth slumps by printing more money. Bua et al. (2014) explain that printing money might fuel inflationary tendencies in a country. In theory, the seignorage government revenue obtained from printing money can become an important means of deficit financing in the short-run. As noted by Guerrero (2006), the average rate of inflation in Zambia is 82.79%, in Sierra Leone is 27.55%, in Burundi, is 10.36%, in The Gambia is 8.72%, and in Egypt is 7.68%.

An observation by Nkurunziza (2005), is that economic instability in Burundi in the 1990s led to a decline in government revenue, which induced the government to rely more on inflation tax. Burundi, which has an unstable government, may prefer high inflation. In turn, Nkurunziza (2001), emphasize that high inflation may signal poor policies and institutions, which cannot implement inflation-reducing policies, such as inflation-forecast targeting policy rules. A study by Boyd et al. (2001) summarizes the causes of high inflation in Burundi that were attributed to an increase in food prices. As a result, inflation rose from an average of 5% annually in the 1970s and 1980s to 16% in the 1990s. Another study by Cukierman et al. (1992), shows that the increase in the price of imported goods prompted the government to devalue its official exchange rate at the time.

In this case, the drawing down of foreign exchange reserves increased the sovereign risk premium. On the other hand, Deme and Fayissa (1995) point out that in the early 1980s in Egypt, inflation steadily rose, reaching an annual growth rate of 14.6%.

According to a monetarist approach, a rise in the growth of the money supply positively influences the domestic inflation for Egypt. This suggests that monetary policy can be applied to control inflation in Egypt. However, Bruno and Easterly (1998), emphasize that a country is considered to be in an inflation crisis if it is above the 40% threshold level of inflation. At this point, growth turns sharply negative, until the end of the inflation crisis period. Countries that had hyperinflation like Zimbabwe experienced negative economic growth. As suggested by Hegerty (2016), sub-Saharan Africa should improve its macroeconomic performance by adopting inflation targeting. In practice, however, most African economies are still using monetary aggregate targeting, and only a few countries like Ghana and South Africa have shifted to rule-based inflation-forecast targeting.

Another study by Kallon (1994), found high inflationary tendencies in Sierra Leone. In particular, the average inflation rate rose from 15.33% in the 1970s to 63% in the 1980s. Despite the fact that the Sierra Leonean economy is an open economy, and relatively small in size, Sierra Leone is still exposed to significant external risks. On the other hand, Odhiambo (2012) investigates the effects of inflation on financial development in Zambia during the period between 1980 and 2011. Since the 1990s, Baldini et al. (2012) points out that Zambian monetary policy focuses on achieving macroeconomic stability through price stability and exchange rate stability. By way of contrast, the country's inflation rate has been noticeably high. This is attributed to deteriorating terms of trade, due to a decline in copper prices during the period 2008 and 2009, exchange rate volatility, high public debt, and supply shocks.

The coefficient on the public debt to GDP ratio is significant at 10% level in Burundi and has a positive sign. This is similar to Teles and Mussolini (2014), who observe that the increase in public debt/GDP ratio can negatively affect economic growth and hence, pushes up the sovereign risk premium. Moreover, Hatchondo et al. (2012) point out that in countries with an undiversified export base such as Burundi, a high public debt/GDP ratio may be interpreted as the outcome of implementing stricter fiscal rules, in order to restore their credibility. Meanwhile, we obtain a significant coefficient on the commodity price in Burundi and the hypothesized negative sign. A rise in commodity prices increases export earnings used to service debt burden, hence, low sovereign risks. This is in line with the findings by Collier (2007). On the other hand, Bruckner and Ciccone (2010) find that a decline in the commodity price worsens the economic conditions in African economies and that as a result, sovereign risks increase. Moreover, Muhanji and Ojah (2011) note that African countries rely on few primary export commodities, which are highly volatile. Further analysis by Senhadji (1997), argues that favorable commodity prices prompt developing countries to over-borrow with the perception that these favorable commodity prices will last a long time. However, commodity prices are usually uncertain, if taking into account circumstances like the oil price shocks of the 1970s. In turn, commodity price booms are short-lived, and as a result, low commodity export earnings lead to high debt accumulation.

The coefficient on GDP growth is significant at 5% and 10% level in Kenya, Nigeria, Zambia, and South Africa and has mixed signs. As explained by Siklos (2011), this implies that an improvement in economic performance may either increase or decrease the sovereign risk premium. Our results show that the improvement in economic growth reduces sovereign risks in Nigeria and South Africa, whereas it doesn't reduce sovereign risks in Zambia and Kenya. Our findings are in line with those of Baek et al. (2005) and Maltritz and Molchanov (2013), who note that a high growth rate reduces the sovereign default risk. Generally, our results confirm that the "resource curse hypothesis" exists in African economies. This means that African economies have a high sovereign risk premium, due to low economic growth, despite the fact that they are richly endowed with the kind of resources that could boost their export earnings. In particular, Auty (2007), defines the "natural resource curse" as the situation where countries endowed with natural resources experience low economic growth. Some

studies, such as those by Sachs and Warner (1999), confirm the negative effects on economic growth. On the contrary, Alexeev and Conrad (2009), disagree.

The coefficient on the foreign exchange reserve is significant in Mauritania, Nigeria, Sierra Leone, and Swaziland. As pointed out by Dreher and Vaubel (2009), the accumulation of the foreign exchange reserve is expected to increase with the economic size and the volume of a country's foreign transactions. Countries with a diversified export base and open economies with high growth rates are expected to hold more reserves. In addition, Bordo et al. (2010) suggest that since foreign currency debt is associated with currency risks, which in turn, push up the sovereign bond spread, holding more reserves might minimize the negative effects of exchange rate risks. Hence, holding an adequate amount of reserves is associated with exchange rate management policies.

The coefficient on the market sentiment is significant at 5% and 10% in Kenya, Mauritania, and Sierra Leone. This is in line with Gonzalez-Rozada and Levy-Yeyati (2008), as well as Bernoth et al. (2012), who find that the increase in the sovereign bond spread is attributed to fluctuations in the investor's risk aversion. High investor's sentiments are interpreted as an increase in the macroeconomic instability, which causes investors to demand a higher risk premium. More recently, Teles and Leme (2010), argue that ratings are used as a guide to investors, where a low sovereign credit rating implies the country carries considerable risk, scaring away potential investors. However, countries with a history of recent defaults have been rated as less risky than countries that have never defaulted. According to Julius et al. (2010), investor's confidence is important when determining the riskiness of a country. Ensuring investors' confidence enhances investors' participation in the market and encourages capital accumulation. On the other hand, Adjasi and Biekpe (2006) find a positive correlation between stock market development and investor confidence in Kenya.

## 7. Conclusions

This study investigates the key drivers of the sovereign risk premium in African countries. The country risk premium is an important factor in determining the choice between foreign currency borrowing and local currency borrowing. As noted by Olabisi and Stein (2015), a high-risk premium in African countries prompts sovereign borrowers to opt for foreign currency borrowing, which exposes them to exchange rate risks and currency mismatches. In particular, Korinek (2011) demonstrates that developing countries are exposed to high sovereign risks due to macroeconomic volatility, which may increase the probability of a sovereign debt crisis. According to the theory of fiscal insurance forwarded by Uribe (2006), under certain monetary and fiscal policy regimes, sovereign risk default is inevitable. From the sovereign risk management perspective, Baldacci and Kumar (2010), indicate that it is, therefore, important for policymakers to control the key drivers that influence a country's risk. However, our study uses a panel-dynamic fixed effects model to identify the key drivers of a country's risk premium.

Our findings indicate that public debt/GDP, GDP growth, inflation rate, foreign exchange reserve, market sentiment, and commodity price are highly significant in influencing the sovereign risk premium. This is consistent with Gonzalez-Rozada and Levy-Yeyati (2008), Ferrucci (2003), Csonto and Ivaschenko (2013), who note that both the fundamental factors and market sentiment influence the sovereign risk premium. In contrast, Martinez et al. (2013) find that the GDP growth and the public debt/GDP ratio are statistically insignificant, although they present the correct sign. Further analysis reveals that the exchange rate, M2/GDP ratio, and trade/GDP ratio are insignificant. On the contrary, Claessens and Qian (1992) and Ebner (2009) find that the exchange rate is highly significant in African countries. Generally, countries with sound sovereign risk management policies and strong fiscal institutions experience lower sovereign risks.

There are a number of important policy implications for African countries. Policymakers should design appropriate sovereign risk management policies oriented towards achieving institutional reforms that will ensure macroeconomic stability is sustainable. For instance, Claessens et al. (2007), claim that the institutional and macroeconomic framework including the monetary, financial, and fiscal

aspects may reflect the level of sovereign risks. In addition, Morgan (2001) suggests that policymakers should set up institutional mechanisms that will manage commodity price uncertainty by issuing commodity-linked bonds and diversifying their export base.

**Author Contributions:** J.M. and C.M. conceived of the presented idea. J.M. developed the theory and derived the model. C.M. supervised all the work. Both authors discussed the results and contributed to the final manuscript.

**Funding:** This research received no external funding.

**Conflicts of Interest:** The authors, Jane Mpapalika and Christopher Malikane declare no conflict of interest.

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
