# Peer review of "The Determinants of Sovereign Risk Premium in African Countries"

_jrfm, doi:10.3390/jrfm12010029_

Round 1

Reviewer 1 Report

This paper is well-written and provides a contribution on the significance of some economic determinants of sovereign risk premia for ten African countries (whose data are available from IMF, World Bank and Federal Reserve Bank of St Louis). It is interesting, even though it deals with a topic which has been studied at length – at least since Cantor and Packer (1996) - and therefore is not very innovative from a methodological point of view. However, it is instructive, because the application refers to some countries which have not been much examined in the literature and the results, although expected, are discussed nicely. The methodology is pretty similar to Martinez et al. , where the application is to Latin-America countries instead of African countries, and is based on simple dynamic fixed effects estimations.

Having come to the end of Section 7, I am not sure whether we got a new and unexpected answer to the question of the main determinants of sovereign risk premia,  so perhaps some comparisons with alternative methodologies is worthy.

Perhaps, some of the comments in this report help to clarify the novelty of the paper and make it more accessible to the JRFM readership.

This paper deals with economic variables only, but, especially for African countries, it is important to include a number of  political and institutional variables as independent variables as well (see also, Catao and Sutton, 2002; Reinhart, Rogoff and Savastano, 2003, among others). Some examples are:

political stability, institutional effectiveness,  risk of armed conflict, risk of social unrest, terrorism threat, international disputes and tensions, quality of bureaucracy, transparency and fairness of legal system, level of corruption, impact of crime, degree of property rights protection, and so on.

I would suggest to compare the methodology employed in this paper with other  recent papers which have studied the problem employing alternative methodologies, e.g., Agliardi, Agliardi, Pinar, Stengos and Topaloglou (2012), where they constructed an optimal country risk index based on stochastic dominance  analysis and singled out various economic, financial and political factors, or Gonzalo and Olmo (2010) (which introduced nonparametric consistent tests of conditional stochastic dominance of arbitrary order in a dynamic setting.

Section 4.3 should be made more concise and put after the description of the dataset.

If inflation –and thus inflation targeting - is a major driver, then a discussion of some event studies where inflation was controlled by policy makers would be helpful.

References

E. AGLIARDI, R. AGLIARDI, M. PINAR, T. STENGOS and N. TOPALAGLOU (2012).  A new country risk index for emerging markets: a stochastic dominance approach, JOURNAL OF EMPIRICAL FINANCE, 19, p. 741-761

CATAO, L., SUTTON, B., 2002. Sovereign Defaults: the Role of Volatility. IMF Working Paper 02/149

GONZALO, J., OLMO J., 2010. Conditional stochastic dominance tests in dynamic settings, Working Paper, we1029, Universidad Carlos III

REINHART, C.M., ROGOFF, K. S., SAVASTANO, M.A., 2003. Debt Intolerance. Brookings Pap. Econ. Act. 1, 1-74

Author Response

I have responded to the Reviewer's comments accordingly. Please find it in the attached file.

Reviewer 2 Report

This is one area there is not much published research. I think the author(s) did a good job. The paper makes a significant contribution to the literature. 

Author Response

I am thankful for this reviewer. He is satisfied with the paper hence no revisions.

Reviewer 3 Report

Line 76.. This study uses quarterly (Needs to be adjusted)

Line 78 Their study finds inflation, terms (Needs to be adjusted)

The authors need to proofread the paper for small typos but the rest is fine

Author Response

I proofread the paper and revised the reviewer's comments as requested.

I am grateful for the positive feedback from this reviewer. The submitted paper is in track changes, you can accept them.
